# The Peptide Functionalized Inorganic Nanoparticles for Cancer-Related Bioanalytical and Biomedical Applications

**DOI:** 10.3390/molecules26113228

**Published:** 2021-05-27

**Authors:** Xiaotong Li, Minghong Jian, Yanhong Sun, Qunyan Zhu, Zhenxin Wang

**Affiliations:** 1State Key Laboratory of Electroanalytical Chemistry, Changchun Institute of Applied Chemistry, Chinese Academy of Sciences, Changchun 130022, China; lixiaotong@ciac.ac.cn (X.L.); mhjian@ciac.ac.cn (M.J.); yhyan@ciac.ac.cn (Y.S.); 2School of Applied Chemistry and Engineering, University of Science and Technology of China, Hefei 230026, China

**Keywords:** peptide ligand, inorganic nanoparticle, cancer, biosensing nanoplatform, nanomedicine

## Abstract

In order to improve their bioapplications, inorganic nanoparticles (NPs) are usually functionalized with specific biomolecules. Peptides with short amino acid sequences have attracted great attention in the NP functionalization since they are easy to be synthesized on a large scale by the automatic synthesizer and can integrate various functionalities including specific biorecognition and therapeutic function into one sequence. Conjugation of peptides with NPs can generate novel theranostic/drug delivery nanosystems with active tumor targeting ability and efficient nanosensing platforms for sensitive detection of various analytes, such as heavy metallic ions and biomarkers. Massive studies demonstrate that applications of the peptide–NP bioconjugates can help to achieve the precise diagnosis and therapy of diseases. In particular, the peptide–NP bioconjugates show tremendous potential for development of effective anti-tumor nanomedicines. This review provides an overview of the effects of properties of peptide functionalized NPs on precise diagnostics and therapy of cancers through summarizing the recent publications on the applications of peptide–NP bioconjugates for biomarkers (antigens and enzymes) and carcinogens (e.g., heavy metallic ions) detection, drug delivery, and imaging-guided therapy. The current challenges and future prospects of the subject are also discussed.

## 1. Introduction

Cancer is listed as the second foremost source of mortality worldwide in 2018 by the report of International Agency for Research on Cancer (IARC) of World Health Organization (WHO) [1]. Diagnostic and therapeutic agents play critical roles in the combat against this malignant disease. Because of their unique physicochemical properties such as large specific surface area, easy functionalization, and excellent optical, electrical, and magnetic properties, a variety of inorganic nanoparticles (NPs) have been extensively studied for early detection and treatment of cancers since 1996 [2,3,4,5,6,7,8,9,10,11,12,13,14,15,16,17,18,19,20,21,22,23,24,25,26,27,28,29,30,31,32,33]. For instance, due to their morphology (size, shape, and structure)-dependent localized surface plasmon resonance (LSPR), colloidal gold NPs (AuNPs) have been employed for the development of simple colorimetric sensing systems for sensitive detection of various cancer-related biomarkers and carcinogens [2,3,4,9,10,11,29]. Rare earth-doped up-conversion NPs (UCNPs), such as NaYF_4_: Yb^3+^, Er^3+^ UCNPs, can be excited by 808 nm and/or 980 near infrared (NIR) lasers, then emit at specific shorter wavelengths, resulting in improved detection specificity through a decrease of the bioluminesce background [6,7,8]. Because noble metal nanoclusters (size less than 2 nm in diameter), carbon dots (CDs, 1 to 10 nm in diameter), and semiconductor quantum dots (QDs) have a large Stokes shift and size-dependent excitation and emission spectra, they can be used as photomedicinal agents for in vitro/in vivo photoluminescence (PL) imaging [6,22,23]. Magnetic nanoparticles such as iron oxide NPs (IONPs) are excellent theranostics for magnetic resonance imaging (MRI)-guided photothermal therapy (PTT) against cancer because they exhibit good biocompatibility, strong magnetic resonance (MR) contrast capacity, and high photothermal conversion efficiency [12,13,14]. In particular, the NPs prefer to accumulate in tumor sites by the size-dependent enhanced permeability and retention (EPR) mechanism [21,28,30,31,32]. This beneficial combination of physical and chemical properties has also given rise to an important application of NPs in the delivery of different anticancer drugs including traditional chemical drugs, small-interfering RNA (siRNA), and antigens [7,16,28,30,31].

Owing to their facile surface chemistry, the accumulation amounts of NPs in tumor site can be further increased through functionalization of NPs with various molecules, which normally enable to specifically recognize the tumor cells and/or tumor neovasculature by ligand-receptor interactions [5,6,9,10,12,13,17,18,19,20,33]. Among of these ligands, peptides with short amino acid sequences have attracted great attention because they have many important physiological functionalities and control nearly all vital functions in humans, including participation in signaling pathways as enzyme substrates, activation of immune defense as antigens, and effect on cellular membrane/organelle membrane functionality as drug carriers or lytic agents [10,12,14,17,18,19,20,33,34,35,36]. Especially, several peptides have been approved as drugs and an increasing number of peptides are entering clinical trials [35,36]. Therefore, a variety of peptide functionalized NPs have been synthesized and applied in bioanalytical and biomedical areas to perform targeting, diagnostic, and therapeutic functions in a single treatment procedure. For example, conjugation of NPs with peptides contained a cyclic RGD motif can generate novel nanosystems which exhibit high tumor-targeting ability through recognition of α_ν_β_3_ integrin receptor on tumor cell surface [37,38]. After being functionalized by cell penetrating peptides (CPPs), AuNP loading doxorubicin (DOX) led to improved survival time of mouse bearing a xenograft intracranial MDA-MB-231 breast tumor because the cellular internalization amount of DOX was increased significantly by the CPP modified AuNPs [39].

As an interactive nanobiotechnological scaffold, peptide functionalized NPs have thoroughly discussed in several reviews, which are categorized either by their components and/or their applications [10,12,14,17,18,19,20,33,34]. However, most of these reviews only described functionalization and application of a single type of NPs. For instance, the preparation and applications (biosensing, diagnosis, and therapy) of peptide modified AuNPs have been summarized in the reviews [10,17,20]. Spicer and colleagues provided a comprehensive overview of the peptide- and protein-functionalized nano-drug delivery vehicles, imaging species, and active therapeutics [14]. Desale and colleagues discussed the impact of CPPs in the field of nanotherapeutics [34]. For obtaining a broader view on the preparation and applications of peptide functionalized NPs, we strongly suggest that audiences read these excellent reviews. In this review, the discussion will focus primarily on the methods of preparing different peptide functionalized NPs and their applications in the cancer-related bioanalytical and biomedical areas including biosensing, bioimaging, drug delivery, and multimodal therapy (as shown in Figure 1). In particular, applications in the areas of cancer diagnosis and tumor-targeting drug delivery therapy are discussed in more detail through highlighted recent publications. Finally, we address future perspectives and the technical challenges of the peptide functionalized NPs as a promising theranostic of cancer.

## 2. Synthesis of Peptide Functionalized Nanoparticles

Generally, the NPs were firstly produced by the hydrothermal or solvothermal approaches [40,41,42,43,44,45,46,47,48,49,50,51,52,53,54,55,56,57,58,59,60,61,62,63,64]. The as-prepared NPs were then modified with different peptides to achieve different functionalities for further applications. There are three main strategies for generating peptide functionalized NPs: (1) ligand exchange, (2) chemical conjugation, and (3) chemical reduction.

### 2.1. Ligand Exchange

The ligand exchange method is the simplest strategy for preparing peptide functionalized NPs, which essentially involves displacement of original ligand on NP surface by a specific peptide ligand and/or a mixture of peptide ligand [40,41]. Cysteine (Cys, C)-containing peptides have been used successfully to synthesize various peptide functionalized AuNPs by ligand exchange method because the thiol group (−HS) of cysteine can form strong S-Au covalent bond with surface Au atoms of AuNPs [40,41,42,43,44,45,46,47,48,49,50,51,52]. As early as 2004, Lévy and colleagues demonstrated that the pentapeptide, CALNN, can convert citrate-capped AuNPs into extremely stable, water-soluble AuNPs with some chemical properties analogous to those of proteins [42]. The CALNN modified AuNPs can be easily further functionalized with other biomolecules (e.g., biotin, DNA, etc.) for biological application. However, the Au−S covalent bond could be decomposed by the thiols (e.g., glutathione (GSH), Cys residues of proteins, etc.) in the living system. In order to eliminate the drawback, Tang and colleagues have developed an approach for synthesizing peptide functionalized AuNPs (peptide−Se−AuNPs) through the Au−Se bond instead of the Au−S bond by using peptide with Se modified cysteine [53,54,55]. The peptide−Se−AuNPs exhibit high colloidal stability, which can resist 5 mmol L^−1^ GSH and achieve a high-fidelity detection. Because phosphate has the ability to react with metallic cations such as Gd^3+^, Fe^3+^, and Zn^2+^ and forms robust metal−phosphate coordination bonds under mild conditions, peptides containing phosphorylated amino acid (e.g., phosphoseryl serine (Ser(P), S(P))) residue have been employed to transfer hydrophobic metallic NPs into aqueous phase and/or functionalize metallic NPs through the formation of metal–phosphate coordination bond [56,57]. For example, Liu and colleagues have developed a simple and robust route for surface functionalization of different NPs with the diameter less than 10 nm including NaGdF_4_ nanodots, IONPs, zinc oxide NPs (ZnONP), AuNPs, and silver NPs (AgNPs) by using tryptone as phase transfer agent [56]. The tryptone is a kind of peptide mixture which contains ca. 10–20% casein phosphopeptide with the sequence S(P)S(P)S(P)EE. The ligand exchange method is normally taken place under mild reaction condition, and can generate peptide functionalized NPs with high colloidal stability and diverse functionality.

### 2.2. Chemical Conjugation

Chemical conjugation a is two-step strategy to attach desired peptides on the NP surfaces. The NPs were firstly capped by stabilizers, such as derivatives of PEG (poly(ethylene glycol)) through ligand exchange and/or physical interactions (e.g., electrostatic interaction, hydrogen bonding, etc.), or hydrophilic shells (e.g., silica (SiO_2_) and/or polydopamine (PDA) shell) through water-in-oil microemulsion method, which often have active groups that can be used to bind peptides [58,59,60,61,62,63,64,65]. The peptides were then conjugated on the NP surface through reaction with the stabilizers. The strategy is very useful to immobilized positively charged/neutral peptides on citrate-capped AuNPs, and transfer hydrophobic NPs into aqueous solution as well as peptide functionalization. In addition, the surface density of peptide on NPs can be adjusted by experimental parameters, such as reaction time and ratio of reagent in the reaction mixture, when the two-step strategy is employed to synthesize peptide functionalized NPs. Using EDC/sulfo-NHS (1-ethyl-3-(3-dimethylaminopropyl) carbodiimide hydrochloride/N-hydroxy sulfosuccinimide) coupling method, Bartczak and Kanaras successfully conjugated the positively charged peptide KPQPRPLS to carboxy-terminated oligoethyleneglycol stabilized AuNPs (OEG NPs, as shown in Figure 2) [58]. Recently, the monocyclic peptide (MCP, the CXC chemokine receptor 4 (CXCR4) antagonist) functionalized manganese-doped iron oxide NPs (MnIO NPs) were synthesized by Fu and colleagues [62]. Using the bifunctional ligand, DIB-PEG-NH_2_ (3,4-dihydroxy benzyl amine-PEG-NH_2_), as a phase transfer agent, the hydrophobic oleate-capped MnIO NPs were transferred into aqueous solution through formation of Mn^2+^/Fe^3+^−DIB chelates. The MCP functionalized MnIO NPs were then synthesized through the amidation reaction between the amine group of PEG and carboxy group of peptide. Li and colleagues employed the carboxyl-terminated SiO_2_ shell for transferring hydrophobic UCNPs from the organic phase to the aqueous phase, and conjugating peptide ligand with high tumor-targeting affinity [64].

### 2.3. Chemical Reduction

The peptide functionalized NPs can also be directly synthesized through the chemical reduction method [66,67,68,69,70,71]. The general operation steps of chemical reduction method are as follows: (1) pre-mixing the metal ion precursor and peptide in reaction solution, (2) adding a small quantity of reducing agent if required, (3) purifying as-prepared peptide functionalized NPs. The whole synthesis process is simple and normally carried out under mild aqueous conditions. In addition, the morphologies of NPs could be modulated by the change of reaction conditions, such as reaction time, pH value, peptide sequence, and ratio of metal ion precursor with peptide. In this strategy, the peptide is responsible for reduction of metal ions as well as stabilization of produced NPs [66,67,68]. Normally, the amino acid residues in peptides such as tyrosine (Tyr, Y), C, aldehyde-functionalized proline (Pro, P), and tryptophan (Trp, W) can reduce the metal ions to correspondent metals through electron transfer [66,67,68]. Si and Mandal reported an approach to prepare tripeptide functionalized AuNPs and AgNPs though an in situ reduction of HAuCl_4_ or AgNO_3_ by a W residue at the C-terminus of peptides (NH_2_-L-Aib-W-OMe and tert-butyloxycarbonyl (Boc)-L-Aib-W-OH) at pH = 11 [66]. In addition, the peptide just plays the role of stabilizing agent, while other chemicals (e.g., such as sodium borohydride (NaBH_4_) and ascorbic acid) are employed as the reducing agents [70,71]. In the presence of NaBH_4_, Corra and colleagues found that the peptide H-H-dL-dD-NH_2_ can be used as capping agent for the straightforward formation of PdNPs, PtNPs, and AuNPs with high monodispersity and colloidal stability in aqueous solution [71].

## 3. Biosensing Platforms Based on Peptide Functionalized Nanoparticles

The peptide functionalized NPs have been extensively employed to construct biosensors/assays with various detection principles for detection of different analytes (some typical examples are included in Table 1) [72,73,74,75,76,77,78,79,80,81,82,83,84,85,86,87,88,89,90,91,92,93,94,95,96,97,98,99,100,101,102,103,104,105,106,107,108,109,110,111,112,113,114,115,116,117,118,119,120,121,122,123,124,125,126,127,128,129,130,131]. Among of these biosensors/assays, colorimetric assays and fluorescence sensing systems were well developed because of excellent optical properties of NPs.

### 3.1. Colorimetric Assays Based on Peptide Functionalized AuNPs

The AuNP has been demonstrated as an excellent nanoprobe of colorimetric assays because its LSPR has an extremely sensitive response towards the dispersion state of AuNP [2,3,4,6,9,10,11]. The LSPR of AuNP displays a red shift along with a visual color change from red to blue, while the dispersion state of AuNP is changed from monodispersing state to aggregating state. Various colorimetric assays based on peptide functionalized AuNPs have been developed for detection of a wide range of cancer-related species such as metallic ions [72,73,74,75,76,77,78,79], small molecules [80], antigens/proteins [81,82,83,84], and enzymes [85,86,87,88,89,90,91,92,93,94,95,96,97,98].

Several colorimetric assays based on peptide functionalized AuNPs have been developed for detection of heavy metal ions since Si and colleagues employed the peptide (sequence, NH_2_-L-Aib-Y-OMe) functionalized AuNPs-based colorimetric assay for sensing mercury ion (Hg^2+^) at 2007 [72]. For instance, Yu and colleagues developed a GSH functionalized AuNPs (GSH-AuNPs) colorimetric assay for on-site detection of Pb^2+^ leaking from lead halide perovskite solar cells (PSCs) [76]. The Pb^2+^-induced aggregation of GSH-AuNPs can read by both naked eye and UV–visible spectroscopy with detection limits (LODs) of 15 and 13 nmol L^−1^, respectively. As shown in Figure 3, Parnsubsakul and colleagues reported a colorimetric assay based on zwitterionic polypeptide, EKEKEKPPPPC ((EK)_3_), capped 40 nm AuNP (termed as, AuNP-(EK)_3_) for sensing nickel ions (Ni^2+^) [79]. By taking advantage of the alternate carboxylic (-COOH)/amine (-NH_2_) groups, the zwitterionic peptide can function dually by being able to sense Ni^2+^ and maintain colloidal stability of AuNPs. Because the aggregation of AuNP-(EK)_3_ can be triggered by Ni^2+^ through interaction of the -NH_2_ group of glutamic acid at the N-terminus of the peptide and Ni^2+^, the color of AuNP-(EK)_3_ solution is changed from red to purple (as shown in Figure 3a). The AuNP-(EK)_3_-based colorimetric assay can detect Ni^2+^ as low as 34 nM within 15 min with a linear range of 60–160 nM (as shown in Figure 3b,c) with high selectivity (as shown in Figure 3d). In addition, AuNP-(EK)_3_-based colorimetric assay can be employed for detection of Ni^2+^ in soil, urine, and water samples since the internal -COOH/-NH_2_ groups of glutamic acid and lysine confer stability to the AuNP-(EK)_3_.

It is found that abnormal enzyme activity is closely related to tumorigenesis and development, making enzymes such as kinases, proteases, and peptidases as important biomarkers for early cancer diagnosis and targets for therapeutic drug development. The colorimetric assays based on peptide functionalized AuNPs can be employed to determine enzyme activity and act as a screening inhibitor of the enzyme, when the AuNPs are functionalized by peptide substrate of enzyme [85,86,87,88,89,90,91,92,93,94,95,96,97,98]. As early as 2006, Wang and colleagues demonstrated that the interactions of biotinylated peptide substrate functionalized AuNPs and avidin-modified AuNPs could be employed to develop a colorimetric assay for the evaluation of kinase activity and inhibition [85]. In this case, using γ-biotin-ATP as a cosubstrate, the kinase reaction results in the biotinylation of the peptide substrate on AuNPs. Mao and colleagues proposed a one-pot and one-step colorimetric sensing method for detecting activity of aminopeptidase N (APN) based on a peptide (NH_2_-FGGFELLC-Ac) functionalized AuNPs/cucurbit[8]uril (pep-AuNPs/CB[8]) supramolecular structure which was formed by the crosslinking of pep-AuNPs with CB[8] [95]. In the presence of APN, the pep-AuNPs/CB[8] supramolecular structure is disassembled because the peptide will be hydrolyzed by APN. The activity of APN can be determined through the absorbance changes based on the assembly/disassembly of AuNPs. Under optimized conditions, the as-proposed colorimetric assay has a linear range from 5 μg/mL to 15 μg/mL with a LOD of 0.42 μg/mL, which can be used to detect APN in serum samples. As shown in Figure 4a,b, Goyal and colleagues reported a heterogeneous protease assay on polyvinylidene fluoride (PVDF) membrane based on aggregation of peptide-functionalized AuNPs [98]. Using the matrix metalloproteinase-7 (MMP-7) substrate (AIEALEKHLEAKGPCDAAQLEKQLEQAFEAFERAG) functionalized AuNPs as typical example, the proteolysis-driven aggregation of AuNPs on the membrane yields a colorimetric response from reddish/brownish to violet with increasing concentration of MMP-7 (as shown in Figure 4c). The color change can be distinguished by the naked eye for MMP-7 concentrations above 165 nM (visual LOD), which is ~4 times lower than that of the same assay performed in homogeneous solution. The practicability of as-proposed assay was demonstrated by detection of MMP-7 in synthetic urine. The colorimetric based on the proteolysis-driven aggregation of AuNPs on PVDF membrane could be used to detect other proteases by using AuNPs functionalized with specific peptides. The proposed approach would be ideal for applications in resource-limited settings.

In addition, peptide-functionalized AuNPs can be used as artificial enzymes (nanozymes) for developed colorimetric assays [99,100]. Feng and colleagues established a colorimetric assay for detection of cellular GSH level by the inhibition effect of GSH on the peroxidase-like activity of GSH stabilized AuNCs (GSH-AuNC) [100]. Because the GSH-AuNC catalytic oxidation of peroxidase substrate 3,30,5,50-tetramethylbenzidine (TMB) is effectively inhibited by GSH, the absorbance at 652 nm is linearly decreased by increasing GSH concentration within a range from 2 to 25 mM. The as-proposed assay provides a powerful tool for identifying cancer cells since the overall GSH level in cancer cells is much higher than that in normal cells.

### 3.2. Fluorescence Assays Based on Peptide Functionalized NPs

Due to the overlap of maximum fluorescence emission of fluorescent probes with UV–visible absorption spectrum (LSPR band) of AuNPs, there is extreme possibility for development of “turn-on” fluorescence assay by peptide functionalized AuNPs [55,101,102,103,104,105]. Recently, Zhang and colleagues developed a “turn-on” fluorescence assay-based peptide functionalized AuNPs nanosensor for simultaneous detection of multiple posttranslational modification (PTM) enzymes, such as histone deacetylase (HDAC) and protein tyrosine phosphatase 1B (PTP1B) [105]. The AuNPs were functionalized by two biotinylated peptide substrates including (FITC-KGRRPED(Ac)K-biotin) with Lys(Ac) as the acetylation site and (biotin-K(Cy5)HRHPRY(P)G) with Y(P) as the phosphorylation site. The AuNPs exhibit strongly quenching capability on the fluorescence of FITC and Cy5. In the presence of specific enzyme pairs (HDAC/rLys-C endoproteinase and PTP1B/Cytochrome oxidase), FITC and Cy5 fluorescence are gradually recovered during enzymatic digestion of peptides, respectively. The activities of HDAC and PTP1B can be quantitatively determined by the recovery of FITC and Cy5 fluorescence, respectively. The approach has LODs of 28 pM for HDAC and 0.8 pM for PTP1B, and can be further employed for screening inhibitors of PTM enzymes and detecting activities of PTM enzymes in HeLa cells.

With the multiple advantages including robust and high photostability, low toxicity, and deep tissue penetration with minimal autofluorescence background, UCNPs are rapidly emerging as strong contenders for the traditional down conversion-based fluorescence NPs/fluorescence dyes in biosensor construction because UCNPs are able to convert near-infrared (NIR) light into higher-energy and multicolor UV–visible emission light [6,7,8]. Recently, peptide-functionalized UCNP-based fluorescence resonance energy transfer (FRET) systems have been also used to detect various enzymes [106,107,108,109,110,111,112,113,114]. UCNPs are normally synthesized in organic phase for obtaining high fluorescence quantum yield and low nanocrystal defect. Therefore, the UCNPs should be firstly transferred into aqueous medium by coating a hydrophilic layer, such as SiO_2_ and PDA, and then conjugated by peptides. In 2015, Zeng and colleagues reported a facile method for preparation of highly compact and stable biofunctionalized UCNPs through peptide-mediated phase transfer strategy [108]. The peptide-functionalized UCNP can be used for high-sensitive detection of trypsin and in vivo evaluation of apoptosis for chemotherapy efficacy of cancer by the FRET between TAMRA (acceptor) and green UCL of UCNP (donor). Recently, Liu and colleagues have developed a series of peptide functionalized UCNP-based FRET sensing platforms for detecting activities of caspases and evaluating treatment efficiency of anticancer drugs (e.g., cisplatin) [65,113,114]. As shown in Figure 5, they constructed a FRET sensing platform for detection of caspase-9 activity in vitro and in vivo by using UCNP@SiO_2_ core@shell NPs functionalized with a Cy5 labeled peptide containing specific motif LEHD for caspase-9 cleavage [114]. In this case, the red up-conversion luminescence (UCL) emission of UCNP@SiO_2_ is efficiently quenched by Cy5. In the presence of caspase-9, the Cy5 dissociated from UCNP@SiO_2_ surface by enzymatic cleavage of LEHD, resulting in recovery of red UCL emission of UCNP@SiO_2_. It is found that the intracellular caspase-9 activity level of cisplatin-treated MG-63 cells is higher than that of cisplatin-treated SW480, indicating that MG-63 cell is sensitive to the cisplatin treatment.

The peptide-based photoluminescent QDs/metal clusters-based fluorescence sensing systems have also been developed for sensitive and selective detection of enzymes such as proteases because these nanoprobes have unique optical properties including high quantum yields, excellent optical stability, narrow fluorescence FWHM and tunable maximum emission wavelength [115,116,117,118,119,120]. Most of these FRET sensors consist of a QDs/metal cluster donor conjugated with a short-sequence peptide substrate carrying the acceptor (e.g., fluorescent dye). In order to increase the FRET efficiency, several acceptor peptides can be conjugated to the same QD. As early as 2006, Shi employed a rhodamine labeled tetrapeptide RGDC to replace the hydrophobic trioctylphosphine oxide (TOPO) ligand of CdSe/ZnS QD [121]. The rhodamine labeled RGDC functionalized CdSe/ZnS QD was successfully used as sensing platform for sensitive detection of MMP activity through the FRET between rhodamine (acceptor) and CdSe/ZnS QD (donor). Lin and colleagues reported subnanometer photoluminescent gold QDs (GQDs) with a peptide ligand that contains nuclear export signal (NES) sequence, nuclear localization signal (NLS) sequence, and capsase-3 recognition sequence (DEVD), which could be used as molecular probes for the real-time monitoring of cellular apoptosis [123].

## 4. Employing Peptide Functionalized Nanoparticle as Positive Tumor-Targeting Nanomedicines

Over recent years, the research community is acknowledging that the peptides can be used as important ligand/building blocks for designing positive tumor-targeting nanomedicines because of their advantages, including their relatively small size, biocompatibility, easy chemical synthesis and modification, and various biofunctionalities [35,36,132]. In particular, (1) large scale synthesis of peptides can be easily achieved by solid-phase synthesis technology, which presents a convenient and economical option for biomedical applications; (2) peptides have good biocompatibility; (3) many peptides have sequence-dependent bioactivity/functions, such as specifically binding to receptors/antigens on (sub)cellular membranes, high cellular permeability, tumor microenvironmental responsiveness, triggering immune response, etc.; (4) the terminus or side chains of peptides provide plenty reactive groups (i.e., amino (-NH_2_) and carboxyl (-COOH) groups) for conjugating others functional molecules, such as drugs or carriers with peptides. With the rapid development of nanotechnology, the peptide-functionalized NPs have been extensively used as nanoprobes and/or nanocarriers for precision oncology [34,38,39,133]. Normally, tumor-specific targeting peptides are used to prepare positive tumor-targeting nanomedicines, which are mainly identified via bio-inspired techniques (biomimetic peptides) or large-scale screening of peptide libraries (such as phage display peptide libraries and chemical peptide libraries) [134,135]. For instance, Wang and colleagues successfully screened two peptide ligands form a one-bead one compound (OBOC) peptide library for the tumor biomarker human epidermal growth factor receptor 2 (HER2) by using in situ single bead sequencing on a microarray [136].

### 4.1. Enhancing Cellular Internalization and Targeting Cancer Cells

Some peptides, known as CPPs, can be uptaken specifically by certain cell organelles through multiple interactions with the exposed plasma membrane [34]. Several CPPs have been identified since the first CPP (peptide with 86 amino acids derived from HIV transactivator of transcription (TAT)) was discovered in 1980 [137]. Generally, CPPs are short-chain amino acid residues (5–30 residues), which are classified into three categories, cationic CPPs, amphipathic CPPs, and hydrophobic CPPs. Cationic CPPs and amphipathic CPPs are normally used to enhance cellular internalization of NPs. After conjugation with CPPs, the cellular internalization of NPs can be enhanced significantly. In particular, the peptides containing special lysine (K)-, arginine (R)-, or proline (P)-rich motifs (known as nuclear localization sequences (NLSs)) can efficiently transport NPs into the nucleus [133,138,139,140,141,142,143,144,145,146,147,148,149]. As early as 1999, Josephson and colleagues found that Tat peptide modified SPIOs were internalized into lymphocytes over 100-fold more efficiently than nonmodified NPs [138]. De la Fuente and Berry found that the Tat-derived CPP (GRKKRRQRRR) functionalized AuNPs exhibited high cell membrane permeability, and were able to target the cell nucleus [141]. In 2008, Sun and colleagues reported that CALNNR_8_/CALNN co-functionalized AuNPs could target different intracellular components through adjusting the ratio of the CALNNR_8_ to CALNN [143]. After incubation with HeLa cells, the peptide mixture (CALNNR_8_:CALNN = 1:9) functionalized AuNPs had translocated into the cell nucleus, while CALNNR_8_ functionalized AuNPs remained in the cytoplasm. Recently, Wang and colleagues demonstrated that Tat (Ac-YGRKKRRQRRR) functionalized cosolvent-bare hydrophobic QD (cS-bQDs-Tat) exhibited extraordinary intracellular targeting performance with the nucleus as the model target [133].

Because RGD motif can bind to integrins with high specificity and affinity and integrin α_v_β_3_ is highly overexpressed in many types of cancer cells, the RGD peptide family have been employed as peptide ligand resources for preparation of peptide-functionalized NPs with positive tumor-targeting capacity [150,151,152,153,154,155,156,157,158,159,160,161,162,163,164]. Among the RGD peptide family, the tumor-targeting abilities of cyclic RGD peptides (such as c(RGDfV) are better than those of linear RGD peptides. As early as 2008, Shukla and colleagues prepared the fluorescein isothiocyanate (FI) and cRGDyK peptide functionalized G5 poly(amidoamine) (PAMAM) dendrimer-entrapped 3 nm AuNPs, which exhibited strong tumor microvasculature targeting efficacy [150]. As shown in Figure 6, Yan and colleagues reported a RGD functionalized ultra-small Gd(OF)_3_: Ce, Tb nanocrystals (about 5 nm in diameter) for simultaneously targeted imaging cell cytoplasm and nucleus [154]. The as-prepared RGD@ Gd(OF)_3_: Ce, Tb nanocrystals were successfully used as fluorescence label for imaging simultaneously the cytoplasm and nucleus of living cells including cancer cells and stem cells since they exhibit good water dispersibility, excellent cellular biocompatibility and targeted ability, high photostability, and double emissions (545 nm and 587 nm) with high quantum yield. After RGD@ Gd(OF)_3_: Ce, Tb nanocrystal labeling, the living cells exhibited very high signal-to-noise ratio of fluorescence emissions. Therefore, RGD@ Gd(OF)_3_: Ce, Tb nanocrystals show great promise for cellular and molecular-level bioimaging applications. Tang and colleagues have successfully conjugated cRGDfK ((Arg-Gly-Asp-DPhe-Lys) on Ag_2_S QDs [155]. The cRGDfK modified Ag_2_S QDs can be selectively internalized by tumor cells and accumulated in tumor tissue, respectively. In particular, the cRGDfK-Ag_2_S QDs exhibit an exceptionally high tumor-to-liver uptake ratio and efficient renal excretion ability, demonstrating it as excellent nanoprobe for molecular imaging of diseases and the monitoring of treatment response.

Other peptide ligands, such as the peptides containing PSP/SP motif, anti-Flt1 peptide (CGNQWFI, AF), and Nestin peptides (AQYLNPS, Nes) have also been used to prepare peptide-functionalized NPs with positive tumor-targeting capacity [57,62,64,165,166,167,168,169,170,171,172]. Because they show highly specific recognition of tumor neovasculature compared to normal blood vessels [173], Chen and colleagues prepared a retro-inverso SP peptide (_D_(PRPSPKMGV(p-S)VS), where p-S is a phosphoseryl serine residue, and others are retro-inverso amino acids) and a cytosol-localizing internalization peptide (KVRVRVRV(dP)P(p-T)RVRERVK, where p-T is a phosphoseryl threonine residue, and dP is Dproline) co-modified NaGdF_4_ ND, which shows high tumor-targeting ability, and facilitates renal clearance [57]. The as-prepared peptide functionalized NaGdF_4_ NDs were successfully used as high efficient contrast agent of MRI for in vivo imaging small drug induced orthotopic colorectal tumor (ca., 195 mm^3^ in volume) in mice. Li and colleagues prepared two SP peptides (L-SP5-C peptide (sequence: _L_(CSVSVGMK(Ac)PSPRP-NH_2_) and L-SP5-H peptide (sequence: _L_(HSVSVGMK(Ac)PSPRP-NH_2_)) modified NaErF_4_:Yb@NaGdF_4_:Yb core@shell UCNPs (UCNP@SiO_2_-L-SP5-C and UCNP@SiO_2_-L-SP5-H) [64]. Due to the tumor-targeting affinity of PSP motif in the peptide ligands, both UCNP@SiO_2_-L-SP5-H and UCNP@SiO_2_-L-SP5-C can be employed as positive tumor targeting contrast agents for UCL/T_1_-weighted MR dual-mode imaging. They found that UCNP@SiO_2_-L-SP5-C has relatively high affinity with HCT116 colorectal cancer (CRC) subtype.

### 4.2. Cancer Therapy

#### 4.2.1. Antiangiogenic Therapy

It is demonstrated the tumor growth and metastasis can be suppressed significantly through disrupting the tumor vascularization [174,175,176]. The tumor inhibition abilities antiangiogenic agents can be reinforced when they are loaded on the NPs. Several peptide-functionalized NPs with antiangiogenic properties have been employed for directly inhibiting pathological angiogenesis [177,178,179,180,181]. Very recently, Li and colleagues synthesized an anti-Flt1 peptide (AF, CGNQWFI) functionalized AuNC (ca. 2 nm in diameter) with targeted antiangiogenic property (as shown in Figure 7) [181]. The AF@AuNCs exhibit effectiveness in inhibiting both tube formation and migration of the endothelial cells in vitro since the interaction between vascular endothelial growth factor receptor 1 (VEGFR1) and its ligands could be blocked by AF and the expression of VEGFR2 could be downregulated (as shown in Figure 7a). The in vivo chick embryo chorioallantoic membrane (CAM) experiment and antitumor experiment with a mouse-bearing CAL-27 tumor verify that the antiangiogenesis and tumor inhibition effect of AF@AuNCs is much stronger than that of free AF (as shown in Figure 7b,c).

#### 4.2.2. Photothermal Therapy

NP-based PTT has been demonstrated as a promising technology for eliminating tumors because photothermal conversion NPs have the ability to efficiently convert light energy into heat [9,10,14,17,182,183,184]. According to current mainstream cancer treatments, an ideal PTT nanoplatform should have following three features: (1) strong tumor-targeting capability for initial delivery to the required site, (2) ability to efficiently convert NIR light energy into heat for improving tissue penetration, (3) high contrast capability for facilitating imaging-guided therapy by MRI, Computed Tomography (CT), and/or fluorescence. Peptide functionalized NPs can satisfy these features because the NPs have unique photothermogenic and optical properties and passive tumor-targeting ability through EPR effect, and peptide ligand have positive tumor-targeting ability [62,185,186,187,188,189,190,191,192,193,194,195]. In particular, the peptide-functionalized NPs could be used as nanomedicines for achieving synergistic photothermal therapy and immunotherapy because some peptides have been demonstrated as therapeutic agents which can efficiently kill cancer cells and inhibit tumor growth [62]. Li and colleagues constructed cancer cell nuclear-targeted copper sulfide NPs (CuS@MSN-TAT-RGD NPs) with a significant photothermal effect to completely kill residual cancer cells and prevent local cancer recurrence through the surface modification of RGD and TAT peptides on 40 nm porous silica coated copper sulfide NPs (CuS@MSN NPs) [189]. The CuS@MSN-TAT-RGD NPs exhibited an outstanding targeting effect toward the cancer cells because of the interaction of RGD with α_v_β_3_, and further transport to nucleus due to the surface decoration of TAT peptides. Under the irradiation of 980 nm NIR laser, the CuS@MSN-TAT-RGD NPs rapidly increase the temperature of the nucleus and induce an exhaustive apoptosis of the cancer cells through destroying the genetic substances. Using mouse-bearing HeLa tumor as a model, the in vivo experiments demonstrated that the xenografted HeLa tumor was completely removed after 2 weeks with one PTT treatment of CuS@MSN-TAT-RGD NPs, and there was no recurrence of the cancer. Fu and colleagues constructed a multifunctional NP (MnIOMCP) for positive tumor-targeting T_1_-weighted and T_2_-weighted (T_1_−T_2_) dual-modal MRI-guided bio-PTT through bioconjugation of the CXCR4 antagonist monocyclic peptides (MCP) with MnIO NPs [62]. The MnIOMCP shows both T_1_-weighted and T_2_-weighted MR contrast abilities, reasonable photothermal conversion efficiency under 808 nm NIR laser irradiation, and the strong tumor-targeting and inhibition of cancer cell growth by the interactions of MCP with overexpressed CXCR4 in the tumor. The experimental result of in vivo experiment demonstrates that MnIOMCP can accumulate in MCF-7 tumors as high as ∼15.9% ID g^−1^ at 1 h after intravenous injection into mice with the aid of an external magnetic field (MF), creating the opportunity for complete eradication of the tumor by T_1_−T_2_ dual-modal MRI-guided bio-PTT. The peptide functionalized NPs can be used as theranostic for achieving PTT and optical bioimaging simultaneously under single laser irradiation. For instance, Huang and colleagues synthesized a tumor-homing peptide, LyP-1(CGNKRTRGC) and NIR dye IR780 co-functionalized gold nanoprisms (GNPs/IR780-LyP-1) [193]. Under 660 nm laser irradiation, the as-obtained GNPs/IR780-LyP-1exhibited the strong surface-enhanced resonant Raman scattering (SERS) property and photothermal conversion efficiency, which can serve as an excellent theranostic for SERS imaging-guided PTT of tumor.

#### 4.2.3. Radiotherapy

The peptide functionalized NPs are also being used as positive tumor-targeting radiation dose enhancer in cancer radiotherapy [173,196,197,198,199,200]. Cruje and colleagues found that the RGD peptide (CKKKKKKGGRGDMFG) functionalized AuNPs exhibited higher enhancement of radiotherapy than that of PEGylated AuNPs [196]. In addition, the smaller RGD peptide functionalized AuNPs (14 nm in diameter) have a three-fold therapeutic enhancement as compared to larger RGD peptide functionalized AuNPs (50 nm in diameter) in MDA-MB-231 cells at clinically relevant 6 MV energy. Recently, Hafsi and colleagues explored the possibility of enhancing two modalities of radiotherapy, X-rays and protons, using RGD peptide functionalized magnetosomes (magnetosomes@RGD) [200]. Here, the magnetosomes are bacterial biogenic magnetic NPs naturally coated with a biological membrane. Compared to unmodified magnetosomes, magnetosomes@RGD exhibited remarkably high cellular internalization as well as efficacy of radiotherapy both in vitro and in vivo. The experimental results indicate that the magnetosomes@RGD can be used as tumor radioenhancers for both X-rays and protons. Interestingly, combined to magnetosomes@RGD, proton therapy showed higher killing efficacy than that of X-ray therapy at equivalent dose.

#### 4.2.4. As Tumor Microenvironment Responsive Nanoprobes with Precision Tumor-Targeting

Owing to their high specificity and sensitivity, several peptide functionalized NP-based nanocomposites have been employed as tumor microenvironment responsive theranostics for precision diagnosis and/or imaging-guided therapy of tumors [201,202,203,204,205,206,207]. For instance, Zhao and colleagues constructed the MMPs/pH dual-stimuli-responsive and reversibly activatable nanoprobe for precision tumor-targeting and fluorescence imaging-guided photothermal therapy through asymmetric cyanine and glycosyl functionalized gold nanorods (AuNRs) with MMPs-specific peptide [203]. As shown in Figure 8, Han and colleagues prepared gemcitabine (GEM) derivative (GEM-GFLG-NH_2_) and pH (low) insertion peptide (pHLIP, AEQNPIYWARYADWLFTTPLLLLDLALLVDADEGT) cofunctionalized magnetic NPs (denoted as GEM-MNP-pHLIP) for targeted therapy of pancreatic cancer with GEM [205]. The pHLIP largely increased the binding affinity of the GEM-MNP-pHLIP to PANC-1 cells. The targeted delivery and effective accumulation of GEM-MNP-pHLIP in vivo were confirmed by MRI enhanced by the underlying magnetic NPs (i.e., Fe_3_O_4_ NPs). After disruption of the dense stroma by metformin (MET), the GEM-MNP-pHLIP treatment exhibited a remarkably improved therapeutic efficacy (up to 91.2% growth inhibition ratio over 30 d of treatment) of both PANC-1 subcutaneous and orthotopic tumor mice models. Very recently, Zha and colleagues synthesized dual-Epstein-Barr virus (EBV)-oncoproteins targeting and pH-responsive NaGdF_4_: Yb^3^^+^, Er^3^^+^@NaGdF_4_ UCNPs (termed as (UCNP-P*n*, *n* = 5, 6, and 7) for precision targeting, monitoring, and inhibition of EBV-associated cancer through conjugation of dual-EBNA1 (a latent cellular protein)/LMP1 (a transmembrane protein)-targeting peptide on UCNP [206]. Because of their transmembrane LMP1 targeting ability and the pH responsiveness capacity, the UCNP-P*n* exhibits high cellular internalization in EBV-infected cells, strong UCL signal enhancement upon targeted dual-protein binding, efficacious EBV cancer inhibition in vitro and in vivo.

#### 4.2.5. Employing Peptide Functionalized NPs as Positive Tumor-Targeting Drug Carries

The peptide functionalized NPs have been extensively employed to avoid non-specific delivery-related issues in the diagnosis and therapy of cancer. Especially, the role of CPP functionalized NPs has been explored in the areas, such as molecular imaging and anticancer drug delivery, because CPPs aid in cellular internalization of various NPs without damaging the cell, and the CPP functionalized NPs can be served as cargos for intracellular transport of various species including radioactive isotopes [173,208,209,210], small molecules [208,211,212,213,214,215,216,217,218,219,220,221,222,223,224,225,226,227,228,229,230,231], oligonucleotides [232,233,234,235,236,237,238,239,240,241,242,243,244,245,246,247,248,249,250,251,252,253,254,255,256], and other peptides [63,257].

As early as 2011, Yang and colleagues prepared a multifunctional nanotheranostic based on superparamagnetic iron oxide NPs (SPIOs) for targeted drug delivery and positron emission tomography/MRI (PET/MRI) dual-modality imaging of tumor [208]. The anticancer drug, DOX was loaded onto the PEGylated SPIO nanocarriers via pH-sensitive bonds, and the tumor targeting peptides c(RGDfC and ^64^Cu chelators, macrocyclic 1,4,7-triazacyclononane-N, N′, N″-triacetic acid (NOTA), were conjugated onto the distal ends of the PEG arms. After reaction with ^64^Cu, the nanotheranostic can be used for dual-modality imaging (MRI/PET) guided chemical therapy of tumor. The nanotheranostic exhibits higher levels of cellular uptake and tumor accumulation than those of its RGD-free counterpart. Lee and colleagues synthesized radiolabeled RGD peptide functionalized Er^3+^/Yb^3+^ co-doped NaGdF_4_ UCNPs (^124^I-c(RGDyk)_2_-UCNPs) by conjugation of the dimeric cyclic RGDyk ((cRGDyk)_2_) peptide and PEG on polyacrylic acid-coated Er^3+^/Yb^3+^ co-doped NaGdF_4_ UCNPs and consecutively radiolabeled with ^124^I [209]. The ^124^I-c(RGDyk)_2_-UCNPs have high specificity for α_v_β_3_ integrin over-expressing U87MG tumor cells and mouse-bearing tumor model, which can be used as a PET/MR/UCL contrast agent with tumor angiogenesis-specific targeting properties. Zhao and colleagues synthesized CTX-functionalized Au PENPs by using polyethylenimine (PEI) as a template for conjugating PEG, glioma-specific peptide (chlorotoxin, CTX, RCLCQPGYCKGRGKGGCCDDCKRAMQHDTTFCPMCM MCMPCFTTDHQMARKCDDCCGGKGRGKCYGPQCLCR-NH_2_) and 3-(4-hydroxyphenyl)propionic acid-OSu (HPAO) and synthesizing 5 nm AuNPs [172]. After radiolabeling^131^I via HPAO, the ^131^I-labeled CTX-functionalized Au PENPs (^131^I-Au PENPs-CTX) showed high radiochemical purity and stability, and could be used as a nanoprobe for the positive targeted SPECT/CT imaging and radionuclide therapy of glioma cells in vitro and in vivo in a mouse-bearing subcutaneous tumor. Due to the unique biological properties of CTX, the as-developed ^131^I-Au PENPs-CTX were able to cross the blood–brain barrier (BBB) and specifically target glioma cells in a rat intracranial glioma model.

The peptide functionalized NPs have been extensively employed for targeting delivery chemodrugs from the injection site to their intracellular targets in tumorous cells [208,211,212,213,214,215,216,217,218,219,220,221,222,223,224,225,226,227,228,229,230,231]. Pan and colleagues have prepared monodispersed TAT peptide modified mesoporous silica NPs (MSNs-TAT) with high payload for nuclear-targeted drug delivery [211]. They found that MSNs-TAT with a diameter of 50 nm or smaller can efficiently target the nucleus and deliver the active anticancer drug DOX into the targeted nucleus, resulting in a significant enhancement in the anticancer activity of the drug. Ai and colleagues employed (pHLIP, NH_2_-AAEQNPIYWARYADWLFTTPLLLLDLALLVDADEGTCG-COOH) to functionalize an 808 nm excited UCNP-based nanoplatform (pyropheophorbide (Ppa) loaded PEGylated UCNP) [228]. The pHLIP-functionalized nanoplatform exhibited an efficient 808 nm NIR laser-irradiated photodynamic therapy (PDT) effect in cancer cells, especially under a slightly acidic condition because the targeting properties of pHLIP to cancer cells under acidic conditions favor the entry of the nanoplatform. The in vivo experiment with a mouse-bearing breast tumor demonstrated that UCNPs were largely accumulated in the tumor site, revealing the excellent tumor-targeting properties of the pHLIP-functionalized nanoplatform, which ensures efficient PDT in vivo [223]. Recently, Xiao and colleagues synthesized a nanosystem (cRGD–PLGA–SPIO@DOX) based on cRGD peptide-functionalized poly(lactic-coglycolic acid) (cRGD–PLGA) block copolymer encapsulated DOX and SPIO for MRI-guided cancer therapy (as shown in Figure 9). The cRGD–PLGA–SPIO@DOX exhibited improved antitumor effects and reduced toxicity compared with free DOX treatment and can act as a theranostic agent for real-time therapeutic monitoring [229].

siRNA-based approach is one of the most promising new therapeutic strategies of cancer treatment, wherein siRNA can regulate the behavior of malignant tumor cells through manipulation of key oncogenes that modulate cellular signaling pathways [232]. The critical point of siRNA-based approach is delivery of the siRNA molecules with high selectivity and efficiency into tumor cells. Several peptide functionalized NP-based strategies have been developed for target-oriented delivery of siRNA with enhanced transfection efficiency [233,234,235,236,237,238,239,240,241,242,243,244,245,246,247,248,249,250,251,252,253,254,255,256]. As early as in 2010, Jung and colleagues studied the interactions of thiol-modified RGD tripeptide, thiol-modified HIV-Tat derived peptide, and EGFRvIII-targeting siRNA cofunctionalized CdSe/CdS/ZnS QDs with two brain cancer cells (U87 cells (overexpressing α_v_β_3_) and PC 12 cells) [233]. They found that the RGD tripeptide functionalization led to higher cellular uptake of siRNA-QDs by the U87 cells than that by the PC-12 cells, resulting in significant decrease of the U87 cell viability. Due to excellent fluorescent property of QDs, the multifunctional siRNA-QDs can also be used for dissecting/tracking signaling cascades triggered by inhibiting specific proteins, showing great potential for simultaneous diagnosis, therapy, and prognosis of cancer. Veiseh and colleagues also confirmed that CTX functionalized SPIOs were well suited for targeted delivery of siRNA to brain cancer cells [234]. Due to their unique properties including pore-size tunability, large loading capacity, excellent biocompatibility, and easy surface modifiability, peptide functionalized mesoporous silica NPs (MSNs) have been demonstrated as promising nanocarriers for systemic delivery of siRNAs with high anticancer efficacy [236,237]. Li and colleagues reported a PEI and fusogenic peptide KALA-functionalized magnetic MSNs (M-MSNs)-based siRNA delivery system (M-MSN_siRNA@PEI-KALA) through the encapsulation of siRNA within the mesopores of M-MSNs, followed by the coating of PEI on the external surface of siRNA-loaded M-MSNs and the chemical conjugation of KALA peptides [236]. Using anti-VEGF siRNA as a typical example, they demonstrated that the M-MSN_siRNA@PEI-KALA enabled to release the loaded siRNA into the cytoplasm and inhibit the tumor growth through the suppression of neovascularization in tumors. Zhu’s group synthesized a series of RGDfC peptide functionalized selenium NP (SeNP)-based nanosystems for delivering different siRNAs with improved anticancer efficiency [239,240,241,242,243]. As shown in Figure 10a, the RGDfC-SeNPs exhibited excellent ability to deliver anti-Oct4 siRNA into HepG2 cells with high gene silencing efficacy (about 1.7 times of conventional lipofectamine 2000) [239]. The RGDfC-SeNPs/siRNA was specifically targeted to the HepG2 tumors and inhibited tumor growth significantly because they can selectively induce HepG2 cell apoptosis through silencing the Oct4 gene and arrested HepG2 cell at G2/M phase (as shown in Figure 10b). The RGDfC-SeNPs can also be used to co-delivery of DOX and siRNA [240]. The in vivo biodistribution experiment indicated that RGDfC-SeNPs@DOX/siRNA were capable of selectively accumulating in the tumor site, resulting in a higher anticancer activity than the free DOX, RGDfC-SeNPs@DOX, or RGDfC-SeNPs/siRNA in vitro and in vivo.

The peptide functionalized NPs can also be used to deliver therapeutic peptides for tumor-targeting therapy [63,257]. As shown in Figure 11, Bian and colleagues developed a nanomedicine (termed as AuNp-DPA) based on small AuNPs (<10 nm in diameter) decorated with a D-peptide p53 activator (DPA, TAWYANFEALLR) coupled with polylysine (PLL) and receptor-targeted peptide (RGD^DP^, CRGDKRGDSP) for targeted tumor therapy in vivo [257]. Take the advantages of the EPR effect and RGD targeting, the AuNP-DPA can successfully deliver DPA into cancer cells and specifically accumulate at tumor sites (as shown in Figure 11a). The results of in vitro and in vivo experiments demonstrated that small AuNP functionalized with therapeutic and targeting peptides has great potency for construction of biocompatible anticancer nanomedicine to overcome physiological and cellular barriers for targeted delivery of therapeutic D-peptides and further awaken their antitumor efficacy (as shown in Figure 11b).

## 5. Conclusions and Future Prospects

In summary, the review discussed the recent progress of nanoplatforms based on peptide functionalized NPs for detection of cancer-related species and diagnosis and therapy of cancer. The combination of the advantages of NPs, such as high surface-area-to-volume ratio and unique optical/electrical/magnetic properties, and the structural and functional characteristics of peptides allows the peptide functionalized NP-based nanoplatforms to detect cancer markers with low LODs, and diagnosis and therapy of cancers with high efficiency. Although many excellent nanoplatforms based on peptide functionalized NPs have been developed during last two decades, only a few of them have used in clinical practices. For example, RGD functionalized GNRs for PTT of cancer has, by now, moved to the stage of preclinical testing.

Efficient clinical translation of the results of peptide functionalized NPs research requires continuous studies. For the in vitro detection, the peptide functionalized NP-based nanoplatforms should be performed in large scale of real samples and can be harmonized by the testing conditions. In addition, the features such as simplicity and cost should be considered since the remote areas lack laboratory equipment and well-trained people. For in vivo applications, the biodistribution, long-term toxicity, and pharmacokinetics of peptide functionalized NPs should be clearly addressed, which requires an auspicious understanding of the physicochemical properties of various peptide functionalized NPs at different nano/bio interfaces. For example, the nonbiodegradable NPs with large hydrodynamic size (more than 10 nm) exhibit long blood circulation half-life, which can significantly increase the time window of imaging and efficiency of therapy. The slow hepatobiliary excretion of large NPs also increases the likelihood of toxicity in vivo. In order to solve this problem, it requires deep cooperation of scientists from different fields including biology, medicine, materials, chemistry, toxicology, pharmacy, and etc. to develop accurate standards/protocols for in vivo prediction/analysis of structure-toxicity relevance. The nano/bio interface studies could be accelerated by the use of high throughput screening methods and research achievements of bioinformatics on prediction of hazard and risk potential. Furthermore, the large-scale production and cost of manufacturing processes should be carefully considered, when the peptide functionalized NPs are used as nanomedicines, in particular as the drug delivery systems, for clinical applications. Although the development of peptide functionalized NP-based theranostics will face continued challenges, they hold great promise for combating cancers.

## Figures and Tables

**Figure 1 molecules-26-03228-f001:**
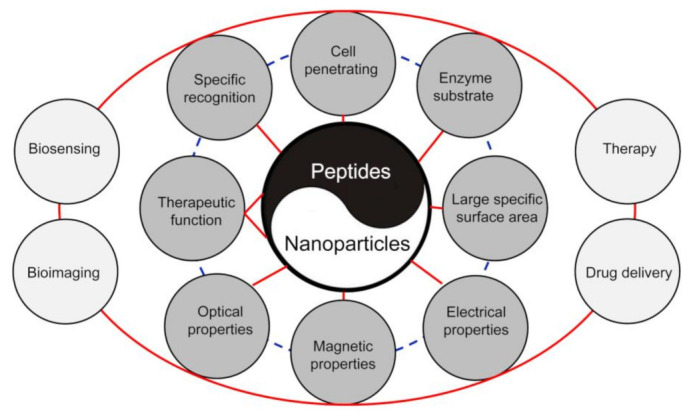
Schematic illustration of the effects of properties of peptide functionalized NPs on bioanalytical and biomedical areas including biosensing, bioimaging, drug delivery, and therapy. Combination of the biological properties of peptides and unique physicochemical properties of NPs can generate excellent nanosensing platforms with high sensitivity and specificity and nanotheranostics with high tumor-targeting capacity.

**Figure 2 molecules-26-03228-f002:**
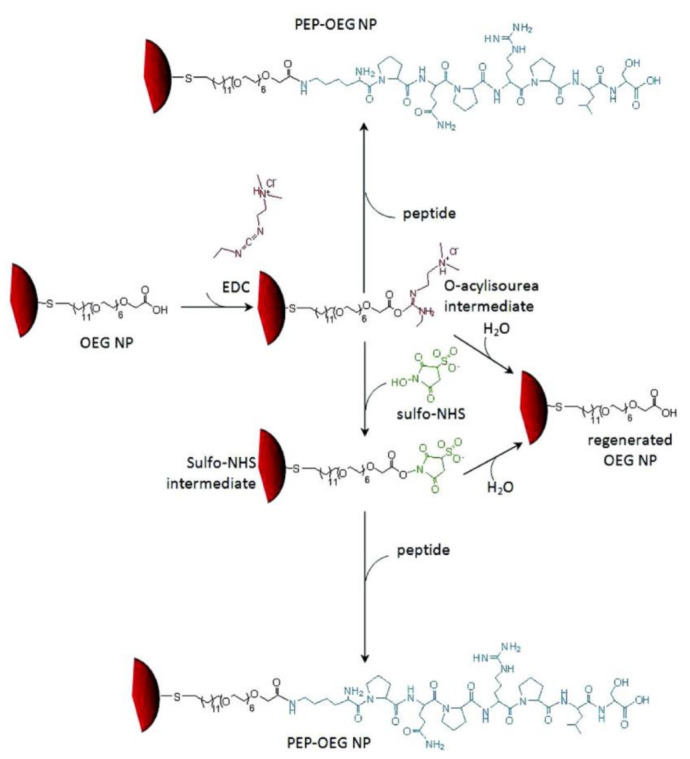
Schematic representation of amide bond formation between the KPQPRPLS peptide and OEG NPs by using EDC and Sulfo-NHS. The degree of peptide coupling as well as the colloidal stability of PEP-OEP NPs are strongly dependent on the experimental conditions (such as EDC and Sulfo-NHS concentrations, peptide concentration, reaction time, and reaction buffer) (Adapted from Bartczak and Kanaras 2011 [58], Copyright 2011 The American Chemical Society and reproduced with permission.).

**Figure 3 molecules-26-03228-f003:**
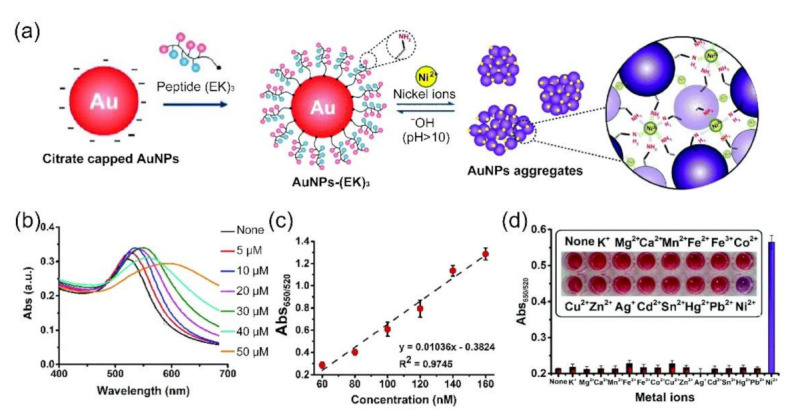
AuNP-(EK)_3_ nanoprobe-based colorimetric detection of Ni^2+^. (**a**) Schematic illustration of AuNP-(EK)_3_ nanoprobe preparation and detection principle of Ni^2+^, (**b**) UV–visible absorption spectra of AuNP-(EK)_3_ at different concentrations of Ni^2+^ (0–50 μM), (**c**) linear calibration plot of AuNP-(EK)_3_ versus Ni^2+^ concentrations in the range of 60–160 nM, (**d**) selectivity of AuNP-(EK)_3_ toward different metal ion species. The zwitterionic region of the (EK)_3_-peptide can bind to Ni^2+^ due to the presence of an -NH_2_ or -COOH group and the unfilled d-orbital of Ni^2+^, leading to the aggregation of AuNPs. As a result, in the presence of Ni^2+^, the color of the AuNP-(EK)_3_ solution is changed from red to purple. (Adapted from Parnsubsakul et al. 2018 [79], Copyright 2018 The Royal Society of Chemistry and reproduced with permission.).

**Figure 4 molecules-26-03228-f004:**
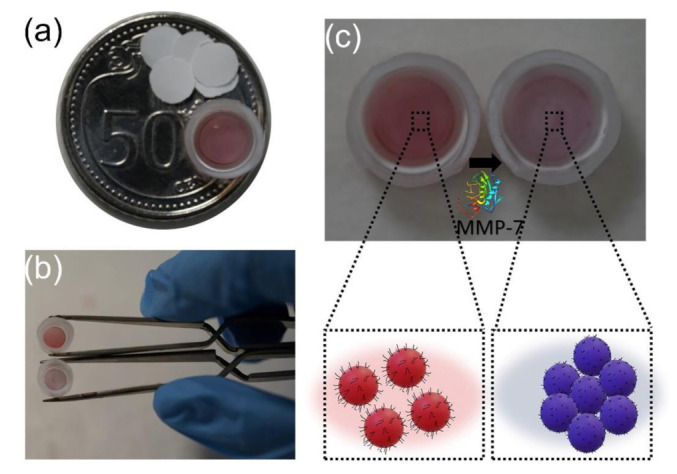
(**a**) Comparison of the size of the membrane with respect to a 50 cents coin, and (**b**) photograph and (**c**) schematics of the assay with peptide functionalized AuNPs on PVDF membrane, which yields a change in color from reddish to violet, due to aggregation induced by MMP-7. The MMP-7 peptide substrate functionalized AuNPs were deposited on PVDF membrane. The AuNPs aggregate on PVDF membrane because of increased electrostatic interparticle attraction upon proteolysis by MMP-7. (Adapted from Goyal et al. 2020 [98], Copyright 2019 Elsevier B.V. and reproduced with permission.).

**Figure 5 molecules-26-03228-f005:**
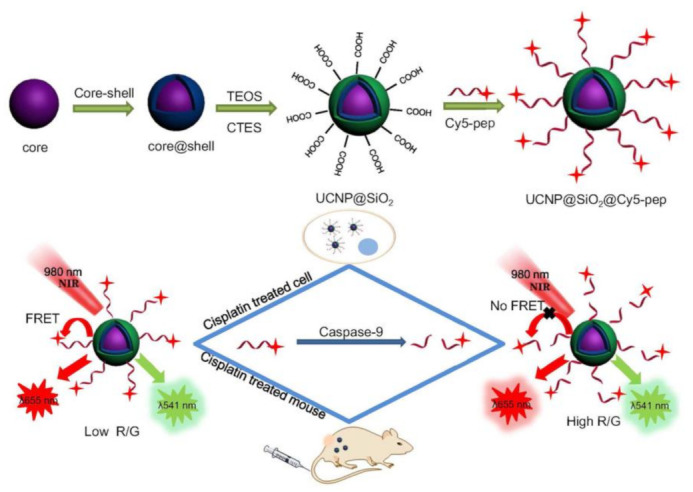
The schematic representation of UCNP-based FRET sensing platform for the detection of caspase-9 activity both in vitro and in vivo by using UCNPs as the energy donor and Cy5 as the energy acceptor. The Cy5 labelled peptide with specific motif LEHD for caspase-9 cleavage were conjugated with carboxyl modified UCNP@SiO_2_ through covalent attachment. (Adapted from Liu et al. 2019 [114], Copyright 2019 Elsevier B.V. and reproduced with permission.).

**Figure 6 molecules-26-03228-f006:**
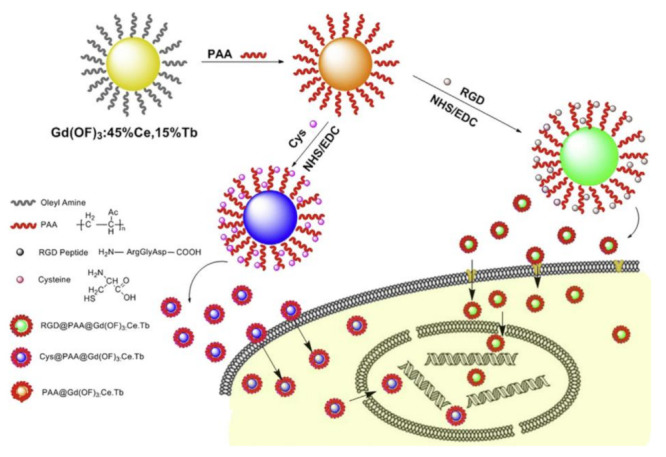
Schematic illustration for the synthesis of RGD and Cys functionalized Gd(OF)_3_: Ce, Tb nanocrystals and their targeted imaging. The oleylamine capped nanocrystals were modified by polyacrylic acid (PAA) to form the PAA capped Gd(OF)_3_: 45%Ce, 15%Tb nanocrystals (PAA@ Gd(OF)_3_: 45%Ce, 15%Tb) through a ligand exchange method. RGD peptide and cysteine were conjugated to the nanocrystals surface by EDC/NHS chemistry (Adapted from Yan et al. 2015 [154], Copyright 2015 Elsevier Ltd. and reproduced with permission.).

**Figure 7 molecules-26-03228-f007:**
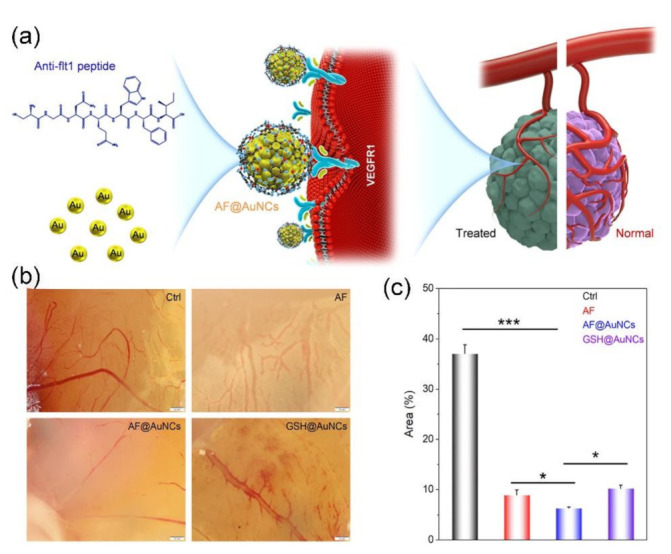
Antiangiogenic cancer therapy by AF@AuNCs. (**a**) Schematic illumination of Anti-Flt1 peptide templated AF@AuNCs for antitumor angiogenesis, (**b**) inhibition of CAM angiogenesis after incubation with AF, AF@AuNCs, and GSH@AuNCs at a concentration of 100 μg/mL and the incubation time was 48 h, and (**c**) comparison of the number of vessels in the AF, AF@AuNCs, and GSH@AuNCs after incubation for 48 h at a concentration of 100 μg/mL in the CAM model. AF@AuNCs were prepared by mixing AF with HAuCl_4_. The AF@AuNCs show enhanced ability in inhibiting angiogenesis in fertilized eggs than those of pure AF and GSH@AuNCs. Data represent means ± SD (*n* = 3). * *p* < 0.05 and *** *p* < 0.001, data with significant difference (Adapted from Li et al. 2021 [181], Copyright 2021 American Chemical Society and reproduced with permission.).

**Figure 8 molecules-26-03228-f008:**
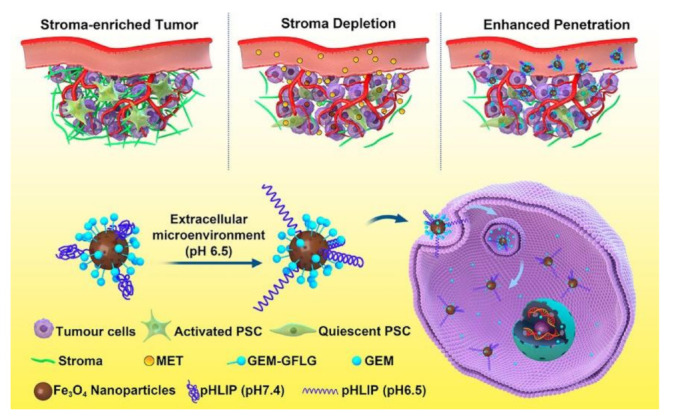
Schematic representation of MET-induced stromal depletion for enhancing the penetration and cathepsin B-triggered release of GEM carried by Fe_3_O_4_ NPs in the lysosome of pancreatic ductal adenocarcinoma (PADC) cells. The pHLIP facilitates the internalization of the underlying Fe_3_O_4_ NPs by inserting into cell membranes because it can format stable transmembrane α-helix in acidic tumor microenvironment. Once internalized into the cancer cells, GEM will be released in lysosome upon cleavage of its linker (i.e., GFLG peptide) by cathepsin B. In the animal experiments, MET was intraperitoneally injected to deplete the dense stromal barrier of PDAC prior to the injection of the above nanoagents to facilitate the effective delivery of GEM. (Adapted from Han et al. 2020 [205], Copyright 2020 American Chemical Society and reproduced with permission.).

**Figure 9 molecules-26-03228-f009:**
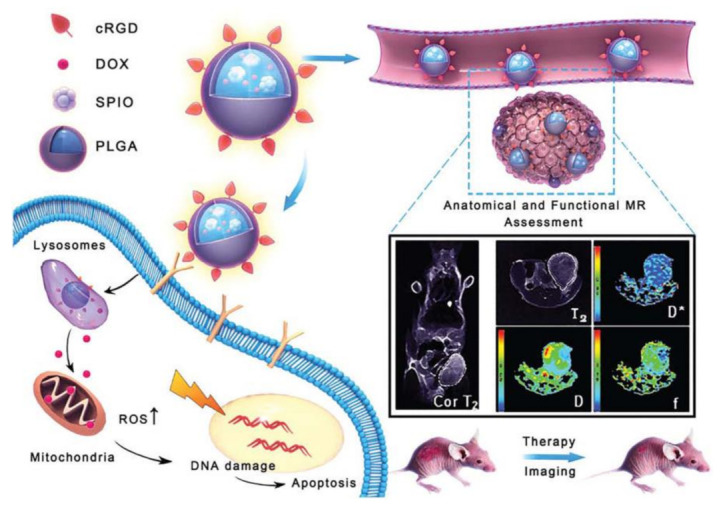
Schematic representation of the cRGD–PLGA–SPIO@DOX multifunctional NPs for targeted tumor therapy and MR imaging. The SPIO and DOX were encapsulated by cRGD peptide-functionalized poly(lactic-coglycolic acid) (cRGD–PLGA) block copolymer. The as-synthesized nanosystem (cRGD–PLGA–SPIO@DOX) exhibited excellent pH-responsive drug release properties under physiological conditions and integrin-targeting ability, which can act as a theranostic agent for MRI-guided cancer therapy (Adapted from Xiao et al. 2019 [229], Copyright 2019 The Royal Society of Chemistry and reproduced with permission.).

**Figure 10 molecules-26-03228-f010:**
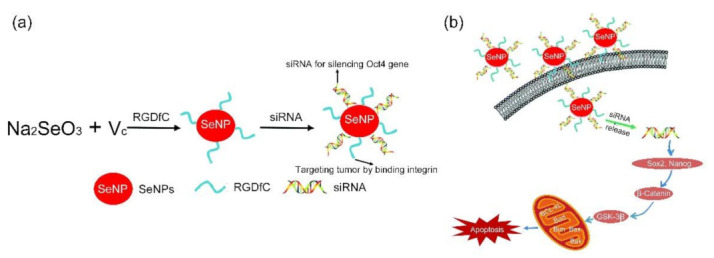
(**a**) Schematic representation of the formation of RGDfC-SeNPs/siRNA, and (**b**) the main signaling pathway of apoptosis induced by RGDfC-SeNPs/siRNA. The SeNPs were synthesized through reduction of Na_2_SeO_3_ by ascorbic acid (Vc), which were functionalized by the cancer-targeting ligand, RGDfC peptide. The siRNA was then loaded on the surface of RGDfC-SeNPs to prepare RGDfC-SeNPs/siRNA. The RGDfC-SeNPs/siRNA can induce cell apoptosis by regulating the Wnt/β-catenin signaling and activation of its downstream target gene (Adapted from Xia et al. 2017 [239], Copyright 2017 The Royal Society of Chemistry and reproduced with permission.).

**Figure 11 molecules-26-03228-f011:**
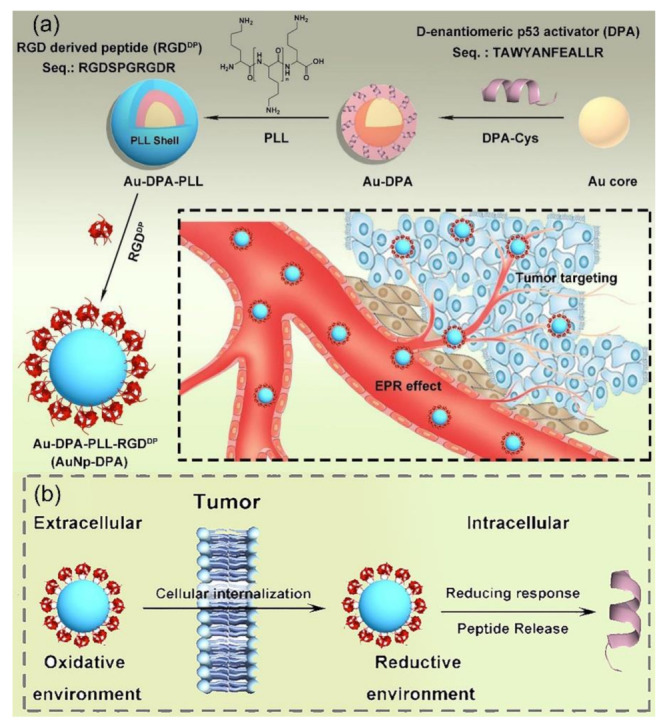
(**a**) Schematic depiction for the synthesis of AuNP-DPA and their enrichment in the tumor site. The AuNPs were modified by the C terminus of DPA (DPA-Cys, encapsulated by PLL, and functionalized by RGD-derived peptide RGDDP. (**b**) The AuNP-DPA can efficiently re-lease DPA to the cytosol through breaking the gold-thiolate bonds by the intracellular reductant, such as GSH. (Adapted from Bian et al. 2018 [257], Copyright 2018 Ivyspring International Pub-lisher and reproduced with permission.).

**Table 1 molecules-26-03228-t001:** Peptide functionalized NP-based biosensors/assays with various detection principles for detection of different analytes. Normally, the peptide functionalized AuNPs were used to developed colorimetric assays, while peptide functionalized fluorescence NPs, such as UCNPs and QDs, were used to construct fluorescence sensing systems.

Nanoparticle	Peptide Sequence	Functionalization Strategy	Analyte	Detection Principle	Linear Range/ Detection Limit	Refs
AuNPs	NH_2_-L-Aib-Y-OMe	Chemical reduction	Hg^2+^	colorimetric assay	4 to 10 ppm/4 ppm	[72]
AuNPs	CALNN	Ligand exchange	Al^3+^	colorimetric assay	0.5 to 6 mM/0.2 mM	[73]
AuNPs	CALNN/GSH	Ligand exchange	Pb^2+^	colorimetric assay	500 nM to 15 mM/500 nM	[74]
AuNPs	GIRLRLEEIEYELKRISGGGC	Ligand exchange	Cu^2+^	colorimetric assay	10 to 150 mM/1 mM	[75]
AuNPs	GSH	Ligand exchange	Pb^2+^	colorimetric assay	30 nM to 2 mM/13 nM	[76]
AuNPs	RFPRGGDD	Ligand exchange	Ag+	colorimetric assay	10 nM to 1 mM/7.4 nM	[77]
AuNPs	EKEKEKPPPPC	Ligand exchange	Ni^2+^	colorimetric assay	60 to 160 nM/34 nM	[79]
AuNPs	CALNNGK_(Abscisic Acid)_G	Ligand exchange	Abscisic acid glucose ester	colorimetric assay	5 nM to 10 mM/2.2 nM	[80]
AuNPs	WHSDMEWWYLLGGGGGC	Ligand exchange	Vascular endothelial growth factor receptor 1	colorimetric assay	0.2 to 10 nM/0.2 nM	[81]
AuNPs	KKHHHHHHKK	Ligand exchange	Prostate specific membrane antigen	colorimetric assay	2 to 10 nM/0.5 nM	[82]
AuNPs	Peptide-p53 and peptide-p14	Ligand exchange	Mdm2	colorimetric assay	30 to 50 nM/20 nM	[83]
AuNPs	H6GLRRAS(P)LG	Chemical conjugation	protein phosphatase 2A	colorimetric assay	-/-	[91]
AuNPs	GPDC or VP-ethylene diamine-DC	Ligand exchange	Dipeptidyl peptidase IV	colorimetric assay	0 to 12 U L^−1^/1.2 U L^−1^ or 0 to 30 U L^−1^/1.5 U L^−1^	[94]
AuNPs	FGGFELLC	Ligand exchange	Aminopeptidase N	colorimetric assay	5 to 15 mg mL^−1^/0.42 mg mL^−1^	[95]
AuNPs	NAADLEKAIEALEKHLEAKGPCDAAQLEKQLEQAFEAFERAG	Ligand exchange	MMP-7	colorimetric assay	5 to 25 mg mL^−1^/3.3 mg mL^−1^	[98]
AuNPs	CCYKKKKQAGDV	Ligand exchange	Integrin GPIIb/IIIa	colorimetric assay	31.25 to 375 ng mL L^−1^/31.25 ng mL L^−1^	[99]
AuNCs	GSH	Chemical reduction	Cancer cell	colorimetric assay	-/-	[100]
AuNPs	FITC-KGRRPED(Ac)K-biotin and biotin-K(Cy5)HRHPRY(P)G	Ligand exchange	histone deacetylase and protein tyrosine phosphatase 1B	FRET	1 nM to 1 mM/28 pM and 0.015 to 0.3 nM/0.8 pM	[105]
UCNPs and carbon NPs	GHHYYGPLGVRGC	Chemical conjugation	MMP-2	FRET	10 to 500 pg mL^−1^/10 pg mL^−1^	[106]
UCNPs	(H)6YGKAGK-TAMRA	Ligand exchange	Trypsin	FRET	0.5−500 nM/0.05 nM	[108]
UCNPs and AuNPs	DDDDARC	Chemical conjugation and ligand exchange	Trypsin	FRET	12 to 208 ng mL^−1^/4.15 ng mL^−1^	[109]
UCNPs	CGRGGLEHDGGRK-Cy5	Chemical conjugation	Caspase-9	FRET	0.5–100 U mL^−1^/0.068 U mL^−1^	[114]
CdSe/ZnS QDs	Rhodamine-RGDC	Ligand exchange	Collagenase	FRET	0 to 5 mg mL^−1^/-	[121]
Gold QDs	NES-linker-DEVD-linker-NLS	Chemical conjugation	Caspase-3	Fluorescence assay	-/-	[123]
Gold nanostars	LRRASLG	Chemical conjugation and ligand exchange	Protein kinase A	Surface-enhanced Raman spectroscopy	5 mU mL^−1^ to 5 kU mL^−1^/5 mU mL^−1^	[125]
AuNPs	3-mercaptopropionic acid-HSSKLQ-K (biotin)	Ligand exchange	Proteolytically active prostate specific antigen	Electrochemical sensor	0.1 to 100 ng mL^−1^/27 pg mL^−1^	[128]
AuNPs	RRRRRAGGPAC	Ligand exchange	Type IV collagenase	Quartz crystal microbalance biosensor	10 to 60 ng mL^−1^/0.96 ng mL^−1^	[131]

## Data Availability

Not applicable.

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
