# Peer review of "The Peptide Functionalized Inorganic Nanoparticles for Cancer-Related Bioanalytical and Biomedical Applications"

_molecules, 2021, doi:10.3390/molecules26113228_

Round 1
Reviewer 1 Report
Xiaotong Li et al. presents a systematic review of peptide-functionalized nanoparticles and their role as diagnostic tools and therapeutic agents. In spite of briefly describing the strategies for obtaining them and some examples of applications, most of the work focuses on the diagnosis and therapy of cancer. The authors are congratulated on addressing this topic of considerable interest. There are some points of criticism I have to make. Detailed comments list below.
- The title does not exactly describe the focus of the present work. A large portion of the systematic review addresses works that illustrate the possible applications of bioconjugates in cancer research, diagnosis and treatment. The other applications are not explored in depth.
- The introduction is a bit confusing and can be improved. Considering the different applications of peptide-functionalized nanoparticles, the authors should give a more holistic and broader view before focusing on aspects related to cancer. Except, if the manuscprit will target this specific area as seen in most of the examples detailed in this manuscript. However, the other sections must be reviewed if this is objective. In this sense, the aspects related to synthesis and biosensors should align with this objective.
- All figures lack a brief caption or description to facilitate their understanding. Additionally, the titles must be adapted to the context of the present review.
- It is suggested to add figures after they are mentioned in the text. For example figure 1.
- The title of figure 1 is not clear. What are the illustrated effects?
- Despite efforts to describe the synthetic strategies of peptide functionalized nanoparticles, this section is little explored. Only one paragraph of each approach has been described. This topic can be enriched. What are the advantages and disadvantages of each? Which is the most used? It would be interesting to make a comparison between these methods.
- Table 1 should have a caption. In addition to the title, a short caption is useful for the reader.
- The authors mentioned focusing on recent publications (line 22). This term is a little vague. For instance, the table 1 considers articles published in 2006. What was the date range used in this review? How were the works covered in this review selected?
- Immobilizing peptides on nanoparticles involves interesting challenges, which are dependent on the physicochemical properties of the components of the conjugates. What are the main characteristics of peptides (net charge, amino acid composition, size and secondary structure) commonly used to build conjugates? Are there any restrictions on size, load or any structure that is not recommended?
- The authors describde that peptide functionalized NPs have used in clinical practices. However, it remains to exemplify and describe the applications of peptide functionalized NPs currently available on the market. It is suggested to the authors to add them in a table.
Minor points:
- Line 61. effect on membrane activity …This is confusing.
- Line 90. A reference is missing.
- Line 424. Antiangiogenic therapy
Author Response
(1) The title does not exactly describe the focus of the present work. A large portion of the systematic review addresses works that illustrate the possible applications of bioconjugates in cancer research, diagnosis and treatment. The other applications are not explored in depth.
Thank you for your suggestion. The title of this manuscript was revised as ‘The Peptide Functionalized Inorganic Nanoparticles for Cancer-Related Bioanalytical and Biomedical Applications’.
(2) The introduction is a bit confusing and can be improved. Considering the different applications of peptide-functionalized nanoparticles, the authors should give a more holistic and broader view before focusing on aspects related to cancer. Except, if the manuscprit will target this specific area as seen in most of the examples detailed in this manuscript. However, the other sections must be reviewed if this is objective. In this sense, the aspects related to synthesis and biosensors should align with this objective.
As an interactive nanobiotechnological scaffold, peptide functionalized NPs have thoroughly discussed in several reviews, which categorized either by their components and/or their applications [10,12,14,17-20,32,33]. However, most of these reviews only described functionalization and application of single type of NPs.
Sentences ‘For instance, the preparation and applications (biosensing, diagnosis and therapy) of peptide modified AuNPs have been summarized in the several reviews [10,17,20]. Spicer and colleagues provided a comprehensive overview of the peptide- and protein-functionalized nano-drug delivery vehicles, imaging species, and active therapeutics [14]. Desale and colleagues discussed the impact of CPPs) in the field of nanotherapeutics [33]. For obtaining broader view on the preparation and applications of peptide functionalized NPs, we strongly suggest that audiences read these excellent reviews.’ for addressing this matter.
(3) All figures lack a brief caption or description to facilitate their understanding. Additionally, the titles must be adapted to the context of the present review.
The brief descriptions were added.
(4) It is suggested to add figures after they are mentioned in the text. For example figure 1.
Due to the required format of Journal, it is difficult to just the figure position and/or size.
(5) The title of figure 1 is not clear. What are the illustrated effects?
The Caption of Figure 1 was re-written.
(6) Despite efforts to describe the synthetic strategies of peptide functionalized nanoparticles, this section is little explored. Only one paragraph of each approach has been described. This topic can be enriched. What are the advantages and disadvantages of each? Which is the most used? It would be interesting to make a comparison between these methods.
There are several reviews on the synthetic strategies of peptide functionalized nanoparticles. The functionalization strategy is defined by the properties of as-synthesized nanoparticles and their applications. There is no the most used method. For instance, AuNPs were normally functionalized by peptides through ligand exchange method and chemical conjugation. UCNPs were normally functionalized by peptides through chemical conjugation.
(7) Table 1 should have a caption. In addition to the title, a short caption is useful for the reader.
A sentence ‘Normally, the peptide functionalized AuNPs were used to developed colorimetric assays, while peptide functionalized fluorescence NPs such as UCNPs and QDs were used to construct fluorescence sensing systems.’ was added for addressing this matter.
(8) The authors mentioned focusing on recent publications (line 22). This term is a little vague. For instance, the table 1 considers articles published in 2006. What was the date range used in this review? How were the works covered in this review selected?
In this manuscript, 259 papers are cited, which includes 180 papers published during 2015 to 2021. Early published papers are cited because they are representative examples on the peptide functionalized NPs synthesis and applications.
(9) Immobilizing peptides on nanoparticles involves interesting challenges, which are dependent on the physicochemical properties of the components of the conjugates. What are the main characteristics of peptides (net charge, amino acid composition, size and secondary structure) commonly used to build conjugates? Are there any restrictions on size, load or any structure that is not recommended?
There is no certain principle/mechanism on the synthesis of peptide functionalized NPs.
(10) The authors describde that peptide functionalized NPs have used in clinical practices. However, it remains to exemplify and describe the applications of peptide functionalized NPs currently available on the market. It is suggested to the authors to add them in a table.
Although many excellent nanoplatforms based on peptide functionalized NPs have been developed during last two decades, only few of them have used in clinical practices (see the end of the first paragraph in the ‘Conclusions and future prospects’ section.
(11) Line 61. effect on membrane activity …This is confusing.
It was revised as ‘and effect on cellular membrane/organelle membrane functionality as drug carriers or lytic agents’.
(12) Line 90. A reference is missing.
References were added.
(13) Line 424. Antiangiogenic therapy
The ‘Antiangiogene therapy’ was revised as ‘Antiangiogenic therapy’.
Reviewer 2 Report
The authors in work entitled: “The peptide functionalized nanoparticles for bioanalytical and biomedical applications”, present extensive review of functionalized nanoparticles by various peptide types. They describe characterization of the nanoparticles, possibility of their application: biosensing, fluorescence assay… Significant part is related to nanoparticles tumor targeting, diagnostics and treatment. The work contains lot of information that are supported with 258 references. Small weakness of the study is only in too many abbreviations that are used, and reader can be lost in text. Maybe, a part could be dedication to abbreviations explanation.
Minor comments:
- Page 2 the first sentence: typo-mistake in “nanoparticles”.
- Resolution of some figures is low and should be increase. Some parts are not readable, e.g. red signs in red backgrounds.
- Page 17 line 639: “As shown as Figure 10”, probably should be in Figure 10.
Author Response
(1) Small weakness of the study is only in too many abbreviations that are used, and reader can be lost in text. Maybe, a part could be dedication to abbreviations explanation.
Thank you for your suggestion. A part has been added at the end of main text for addressing this matter.
(2) Page 2 the first sentence: typo-mistake in “nanoparticles”.
The ‘nanpparticles’ was corrected as ‘nanoparticles’.
(3) Resolution of some figures is low and should be increase. Some parts are not readable, e.g. red signs in red backgrounds.
The quality of figures were improved.
(4) Page 17 line 639: “As shown as Figure 10”, probably should be in Figure 10.
The ‘As shown as Figure 10’ was corrected as ‘As shown in Figure 10’.
Reviewer 3 Report
The manuscript submitted by Li et al. to Molecules is a review on peptides-functionalized nanoparticles for bioanalytical and biomedical applications. Overall, the manuscript needs to be proof-read carefully to correct typos, grammatical errors and English. Moreover, since the described nanoparticles are only inorganic ones, I suggest to the authors to modified their title to include this fact as follows “Peptide Functionalized Inorganic Nanoparticles for Bioanalytical and Biomedical Applications.”. The manuscript is however interesting but I have several questions and remarks that must be addressed before any publication. I therefore recommend to reconsider the manuscript of Li et al. for publication in Molecules after major revision. Please find below specific remarks and/or questions.
1. Page 1, line 44: I propose to the authors to change “served” to “used”.
2. Page 2, line 48: “MR”, the authors have to define all the abbreviations used.
3. Page 2, line50 “EPR effect”: The authors have to give references of manuscripts recently published by Maeda who was the first to describe the EPR effect 35 years ago. Maeda published in 2021 a review on the EPR effect: Maeda, H. J. Personalized Medicine, 2021, 11, 229.
4. Introduction part: The authors have to mention the advantages and drawbacks of inorganic nanoparticles. Moreover, several sentences need to be re-worded.
5. Page 3, line 100: I guess that the word “bound” is missing after “Au-S covalent”.
6. Page 3, lines 103-104: Which kind of other biomolecules, for which applications?
7. Page 3, lines 104-105: How can the Au-S bound be decomposed? Which kind of thiols? The authors have to summarize the Au-S bound decomposition by thiols and give some naturally available thiol compounds able to decompose Au-S bound.
8. Page 3, lines 115-120: And? What are the advantages of such method?
9. Page 3, lines 123-126: How is this first step realized? The authors must briefly describe it.
10. Page 3, lines 127-129: I don’t understand the sentence “The strategy is very useful … as peptide functionalization.”.
11. Page 3, line 131: Which two-step strategy?
12. Page 4, line 142: Where is the reference? What is the modification yield?
13. Page 4, line144: How was the peptide conjugation realized? How are nanoparticles purified after conjugation? How are side-products eliminated?
14. Page 4, lines 159-162: The sentence needs to be re-worded. What is the meaning of the letter “C”?
15. Page 5, lines 167-170: Words are missing in this sentence. In conclusion of this second part, the authors have to compare the three described methods giving their advantages and drawbacks. Which one is the most used to functionalize inorganic NPs?
16. Page 5, Table 1: How were these NPs functionalized? It may be interesting to add a column giving the method used to functionalized the inorganic NPs.
17. Page 7, lines 183-187: And? Are they efficient in detecting cancer-related species? Were the studies conducted in vitro, in vivo?
18. Page 7 lines 188-191: This sentence must be re-worded.
19. Page 7, lines 194-195: I guess that words are missing in this sentence.
20. Page 7, Figure 3: This figure is really too small to be able to read information given in the figure. The authors have to improve its quality.
21. Page 7, lines 220-223: I don’t understand how this system can work.
22. Page 8, line 231: What does “LOD” mean?
23. Page 8, line 256: There is an extra “the” in this sentence.
24. Page 9, line 270: Words are missing.
25. Page 11, TAT and Tat: At the beginning of this page, the authors used TAT, and few lines latter they used Tat, and on page 13 they used again TAT. Which one is the correct way to write the peptide?
26. Page 11, Figure 6: The legend if this figure is really too small and impossible to read. The authors have to improve the quality of Figure 6. Same remark for Figure 7 on page 13
27. Page 13, line 455: The abbreviation “CT” has to be defined.
28. Page 14, lines 478-479: What do the authors mean by “reasonable photothermal conversion”?
29. Page 14, lines 496-497: How is NPs radiolabeled?
30. Page 15, line 553: The reference is not correctly written.
31. Page 16, line 600: “poly(lacticco-glycolic acid)” has to be changed to “poly(lactic-coglycolic acid)”.
32. Page 17, line 638: There is an extra “for delivering”.
33. Page 18, line653: I guess that “therapic” has to be changed to “therapeutic”.
Author Response
(1) I suggest to the authors to modified their title to include this fact as follows “Peptide Functionalized Inorganic Nanoparticles for Bioanalytical and Biomedical Applications.”
The title of the manuscript has been revised as ‘‘The Peptide Functionalized Inorganic Nanoparticles for Cancer-Related Bioanalytical and Biomedical Applications’’.
(2) Page 1, line 44: I propose to the authors to change “served” to “used”.
The “served” was replaced by “used”.
(3) Page 2, line 48: “MR”, the authors have to define all the abbreviations used.
A part has been added at the end of main text for addressing the abbreviations used.
(4) Page 2, line50 “EPR effect”: The authors have to give references of manuscripts recently published by Maeda who was the first to describe the EPR effect 35 years ago. Maeda published in 2021 a review on the EPR effect: Maeda, H. J. Personalized Medicine, 2021, 11, 229.
The mentioned paper was cited (see ref. 32).
(5) Introduction part: The authors have to mention the advantages and drawbacks of inorganic nanoparticles. Moreover, several sentences need to be re-worded.
Thank you for your suggestion. Sentences ‘Because of their unique physicochemical properties such as large specific surface area, easy functionalization, and excellent optical, electrical and magnetic properties,’ (Introduction section) and ‘For example, the nonbiodegradable NPs with large hydrodynamic size (more than 10 nm) exhibit long blood circulation half-life, which can significantly increase the time window of imaging and efficiency of therapy. The slow hepatobiliary excretion of large NPs also increases the likelihood of toxicity in vivo.’ (Conclusions and future prospects section) were added in the revised manuscript for addressing this matter.
(6) Page 3, line 100: I guess that the word “bound” is missing after “Au-S covalent”.
The word ‘bond’ was added after “Au-S covalent”.
(7) Page 3, lines 103-104: Which kind of other biomolecules, for which applications?
‘(e.g., biotin, DNA, etc.)’ was added for addressing this matter.
(8) Page 3, lines 104-105: How can the Au-S bound be decomposed? Which kind of thiols? The authors have to summarize the Au-S bound decomposition by thiols and give some naturally available thiol compounds able to decompose Au-S bound.
‘(e.g., glutathione (GSH), Cys residues of proteins, etc.)’ was added for addressing this matter. The following sentences are also intended to illustrate the problem.
(9) Page 3, lines 115-120: And? What are the advantages of such method?
A sentence ‘The ligand exchange method is normally taken place under mild reaction condition, and can generate peptide functionalized NPs with high colloidal stability and diverse functionality.’ was added for addressing this matter.
(10) Page 3, lines 123-126: How is this first step realized? The authors must briefly describe it.
‘through ligand exchange and/or physical interactions (e.g., electrostatic interaction, hydrogen bonding, etc.)’ and ‘through water-in-oil microemulsion method’ were added for addressing this matter.
(11) Page 3, lines 127-129: I don’t understand the sentence “The strategy is very useful … as peptide functionalization.”.
The positively charged/neutral peptides can cause aggregation of citrate-capped AuNPs if these kind of peptides were directly mixed with citrate-capped AuNPs.
(12) Page 3, line 131: Which two-step strategy?
All of two-step strategies. It is a general advantage of two-step strategy.
(13) Page 4, line 142: Where is the reference? What is the modification yield?
It is ref. 62. The modification yield is nearly 100% because excess of MCP (>10000 times) were used.
(14) Page 4, line 144: How was the peptide conjugation realized? How are nanoparticles purified after conjugation? How are side-products eliminated?
Please read ref.62. The manuscript is a review, which can not copy too many detailed information from published literature.
(15) Page 4, lines 159-162: The sentence needs to be re-worded. What is the meaning of the letter “C”?
The sentence was revised as ‘Normally, the amino acid residues in peptides such as tyrosine (Tyr, Y), C, aldehyde-functionalized proline (Pro, P) and tryptophan (Trp, W) can reduce the metal ions to correspondent metals through electron transfer.’ The ‘C’ is acronym of cysteine, which was explained in section 2.1.
(16) Page 5, lines 167-170: Words are missing in this sentence. In conclusion of this second part, the authors have to compare the three described methods giving their advantages and drawbacks. Which one is the most used to functionalize inorganic NPs?
The sentence was revised as ‘Corra and colleagues found that the peptide H-H-dL-dD-NH2 can be used as capping agent for the straightforward formation of PdNPs, PtNPs, and AuNPs with high monodispersity and colloidal stability in aqueous solution.’
The functionalization strategy is defined by the properties of as-synthesized nanoparticles and their applications. There is no the most used method. For instance, AuNPs were normally functionalized by peptides through ligand exchange method and chemical conjugation. UCNPs were normally functionalized by peptides through chemical conjugation.
(17) Page 5, Table 1: How were these NPs functionalized? It may be interesting to add a column giving the method used to functionalized the inorganic NPs.
A column was added for addressing this matter.
(18) Page 7, lines 183-187: And? Are they efficient in detecting cancer-related species? Were the studies conducted in vitro, in vivo?
The following three paragraphs and Table 1 were shown several typical examples of the colorimetric assays based on peptide functionalized AuNPs. Yes, they are efficient in detecting cancer-related species (please see Table 1). It is well known that colorimetric assay is in vitro detect technique. There is few colorimetric assay can be employed for in vivo detection.
(19) Page 7 lines 188-191: This sentence must be re-worded.
The sentence was revised as ‘Several colorimetric assays based on peptide functionalized AuNPs have been developed for detection of heavy metal ions since Si and colleagues employed the peptide (sequence, NH2−L−Aib−Y−OMe) functionalized AuNPs colorimetric assay for sensing mercury ion (Hg2+) at 2007.’
(20) Page 7, lines 194-195: I guess that words are missing in this sentence.
The sentence was revised as ‘The Pb2+−induced aggregation of GSH−AuNPs can read by both naked eye and UV−visible spectroscopy with detection limits (LODs) of 15 and 13 nmol L-1, respectively.’.
(21) Page 7, Figure 3: This figure is really too small to be able to read information given in the figure. The authors have to improve its quality.
The quality of Figure 3 was improved.
(22) Page 7, lines 220-223: I don’t understand how this system can work.
A sentence ‘In this case, using g-biotin-ATP as a cosubstrate, the kinase reaction results in the biotinylation of the peptide substrate on AuNPs.’ Was added for addressing this matter.
(23) Page 8, line 231: What does “LOD” mean?
The ‘LOD’ means ‘detection limit’, which was explained in section 3.1 (just before ‘As shown in Figure 3’).
(24) Page 8, line 256: There is an extra “the” in this sentence.
The extra “the” was deleted.
(25) Page 9, line 270: Words are missing.
The sentence was revised as ‘The AuNPs exhibit strongly quenching capability on the fluorescence of FITC and Cy5.’.
(26) Page 11, TAT and Tat: At the beginning of this page, the authors used TAT, and few lines latter they used Tat, and on page 13 they used again TAT. Which one is the correct way to write the peptide?
Both of them are correct. Some colleagues like ‘Tat’, for example, refs. 138, 141, 149 and 234, while some colleagues like ‘TAT’, for example refs. 190 and 212. Although we like ‘TAT’, in order to respect originality, we didn't unify the name of this peptide.
(27) Page 11, Figure 6: The legend of this figure is really too small and impossible to read. The authors have to improve the quality of Figure 6. Same remark for Figure 7 on page 13
We try our best to improve the quality of figures. Because the figures are reproduced artworks, which are strongly limited by the quality of the original images.
(28) Page 13, line 455: The abbreviation “CT” has to be defined.
‘Computed Tomography’ was added for addressing this matter.
(29) Page 14, lines 478-479: What do the authors mean by “reasonable photothermal conversion”?
It means that the photothermal conversion efficiencies of NPs can satisfy the in vivo PTT.
(30) Page 14, lines 496-497: How is NPs radiolabeled?
The NPs were not radiolabeled, which were just used as radiotherapy enhancer.
(31) Page 15, line 553: The reference is not correctly written.
The literature number of the reference ‘2002919’ was added.
(32) Page 16, line 600: “poly(lacticco-glycolic acid)” has to be changed to “poly(lactic-coglycolic acid)”.
It was changed in the revised version.
(33) Page 17, line 638: There is an extra “for delivering”.
The extra “for delivering” was deleted.
(34) Page 18, line 653: I guess that “therapic” has to be changed to “therapeutic”.
The “therapeutic” was corrected as “therapeutic”.
Round 2
Reviewer 1 Report
The topic addressed is especially important in peptide research area. The authors have responded to all comments. The manuscript was significantly improved with the modifications based on reviewer´s suggestions. I acknowledge the effort and good work done by the authors, however, I suggest small details that can contribute to the present review.
1. There are abbreviations not specified in the manuscript, such as PADC cells.
2. The title of figure 1 still remains confused. I cannot observe the effects described. Only the properties and applications are being represented in the figure.
3. The authors described: "...Although many excellent nanoplatforms based on peptide functionalized NPs have been developed during last two decades, only few of them have used in clinical practices....". The authors are invited to specify which are peptide functionalized NPs currently available in clinical practice. This will greatly enrich the work, with a more real view of the translation of this technology. Although there are few examples, a brief description of them or a table (if it is posible), would be very informative.
4. The title of some figures exactly matches the figure title of the reference used. I suggest revising the title of each figure, updating the context. I suggest paraphrase and write using your own words. There are figures with components (B and C) that are not explored in the text.
Author Response
Thank you for your comments and suggestions!
- There are abbreviations not specified in the manuscript, such as PADC cells.
The full name ‘pancreatic ductal adenocarcinoma’ was added in the text.
- The title of figure 1 still remains confused. I cannot observe the effects described. Only the properties and applications are being represented in the figure.
The words ‘with high sensitivity and specificity’ and ‘high tumor-targeting capacity’ have been added for addressing this matter.
- The authors described: "...Although many excellent nanoplatforms based on peptide functionalized NPs have been developed during last two decades, only few of them have used in clinical practices....". The authors are invited to specify which are peptide functionalized NPs currently available in clinical practice.This will greatly enrich the work, with a more real view of the translation of this technology.Although there are few examples, a brief description of them or a table (if it is posible), would be very informative.
No peptide functionalized NPs were selected for clinical testing until now. Some of them (e.g., RGD functionalized GNR for PTT of cancer) have by now moved to the stage of preclinical testing. A sentence ‘For example, RGD functionalized GNRs for PTT of cancer has by now moved to the stage of preclinical testing.’ was added for addressing this matter.
- The title of some figures exactly matches the figure title of the reference used.I suggest revising the title of each figure, updating the context. I suggest paraphrase and write using your own words. There are figures with components (B and C) that are not explored in the text.
Part of Figure captions were revised. Figures with components (B and C) were addressed in the text.
Reviewer 3 Report
The authors have addressed all the questions/remarks done by the reviewers.
I do therefore recommend the publication of their manuscript in Molecules.
Author Response
Thank you for your comments and suggestions!